# The Uniformity of Syntactic Structures in Various Natural Languages

## Abstract

A variety of word orders exists, which gave rise to thousands of natural languages around the world. We demonstrate a non-computational method for amalgamating different languages' syntactic structures into the same model per expression. By using a non-linear approach in sequencing words, we uncover what may be the hidden nature of syntactic uniformity that is universal across all natural languages in hopes of introducing a better approach to machine translation.

## 1 Introduction

Words of a sentence can be arranged in several ways for different languages and yet still convey the same meaning. The existence of variety in syntax has served as an indication that natural languages are intrinsically heterogeneous rather than homogeneous. Noam Chomsky, on the other hand, has been a proponent of Universal Grammar (UG), a theory that all natural languages share essentially the same grammar or syntax at a hidden level. (Barman, 2012). If humans are equipped with the language faculty from birth, then first language (L1) acquisition occurs in children the exact same way regardless which languages they are exposed to. A Jamaican child being raised in the U.K. learns English just like other children in the same location for instance. A Caucasian child from Poland will speak Filipino fluently if she grows up in the Philippines. However, non-human primates and other mammals do not appear to possess the same language traits as Homo sapiens do. Numerous scientists made attempts to install language into chimpanzees and gorillas, but not one could utilize language at the level of humans. This likely suggests animals lack the language faculty, which is responsible for combining words together to create discrete sentences and thoughts.

Linguists have used syntax trees or parse trees to represent sentences visually. Syntax trees are used to illustrate sentences as hierarchical structures with lexical categories describing the type of each word. The syntactic structure of a sentence can shape its meaning like the lexical semantics of individual words. Because of this reason, there might not be any difference between *semantics* and *syntax* since both concepts are all part of mental representations. (Chomsky, 2000). However, this conjecture appears to be false as two sentences in two different languages can convey the exact same meaning while having completely different syntactic structures even if the words are exactly the same. Furthermore, hierarchical representations of syntax imply there exists an unknown property of sentence formation, which causes words to be linked together grammatically to create sentences and therefore thoughts. No one has yet been able to demonstrate how such a process can take place almost instantaneously in the human brain. The swiftness of sentence formation suggests language utilization is more of a simple process and not a complex one.

So far, practical applications of syntax trees have been very limited in scope and use. The descriptive nature of syntax trees makes them useful for sentence analysis but not necessarily for sentence formation or translation. Recently, sentence parsing has become more practical for real-world application with the development of Universal Dependencies (UD). Because the UD schema can show how content words such as nouns and verbs within a sentence are related to each other, it can be quite useful for sentence analysis. This is a major improvement over the traditional method of parsing syntax trees. (Kondratyuk et al., 2019).

The recent machine-learning paradigm has drastically improved the performance of natural language processing (NLP). Language models such as BERT (Bidirectional Encoder

Representations from Transformers) and GPT-3 (Generative Pretrained Transformer 3) have the ability to generate sentences and paragraphs that may be indistinguishable from the ones created by humans. The combination of having big data of corpora and the development of research in artificial neural networks has significantly improved the quality of machine translation (MT) over the years. Neural machine translation (NMT) has shown improvements to previous models of statistical machine translation (SMT) by training artificial neural networks to yield better translation results. (Bahdanau et al., 2015). However, the new method still remains as a probabilistic approach. That means getting anywhere near 100% accuracy in translation is highly unrealistic since it has to estimate what the right answer likely is. Although neural machine translation seems to be very promising, it is not without its own set of limitations. (Castilho, 2017).

## 2    Method

We use three samples in five languages to illustrate the uniformity of sentences structures in different word orders. Three out of the six possible word orders will be examined; Subject-Verb-Object (English and French), Subject-Object-Verb (Japanese and Uzbek), and Verb-Subject-Object (Welsh). These word orders make up most of all natural languages in the world, especially SVO and SOV. (Carnie, 2002). The example sentences contain between seven and ten words per sentence to show a level of complexity in their syntactic structures that is neither too simple nor too complex. Sentences with only a few words lack any kind of complexity and may not be substantive for discussion. Sentences that are excessively long or complex are likely too laborious for analysis. However, we will explore one overly complex sentence to showcase the validity and the scope of the method.

## 3    Syntactic Structures

### 3.1    Word Orders

Natural languages have different ways of putting words together to convey meaning. In simple sentences such as *John is sick*, one of six arrangements is used to connect the three constituents: (1) *John is sick*, (2) *John sick is*, (3) *Is sick John*, (4) *Is John sick*, (5) *Sick John is*, and (6) *Sick is John*. These word orders are SVO, SOV, VOS, VSO, OSV, and OVS, respectively (S stands for *subject*, V stands for *verb*, and O stands for *object*). The subject refers to the main entity or concept of the sentence. The verb describes an action or a condition regarding the subject. The object is an entity, concept, or description that is related to the subject. A sentence must have a verb and is required to have a subject; although a subject can be omitted or implied in some circumstances for some languages. An object may or may not be required.

Even though word orders are useful for classifying languages, they do not offer much practicality for machine translation. One way to make word orders more applicable to real-world usage is to replace *subject*, *object*, and *verb* with terminology generally associated with lexical categories and syntax trees such as *noun phrase*, *verb*, and *predicate*. By doing so, it becomes possible to classify every word in a sentence while still maintaining the conceptualization of word orders. Nevertheless, we will continue to make use of the traditional description of word orders whenever we find them useful or applicable.

### 3.2    Types of Sentences

There are several types of sentences that can convey meaning. The most common type is declarative sentences. They are statements that define relationships between different concepts (e.g. bird and tree). In English, a declarative sentence usually starts with a subject or a *primary noun phrase* (PNP) as such as "John" or "Emily's car." Then it is followed by a verb or a *primary verb phrase* (PVP) such as "observe" or "go up." What follows is the rest of the predicate, a declarative sentence minus its subject. Therefore, the word *subpredicate* (SP) can be defined as a declarative sentence excluding PNP and PVP. Then a declarative sentence (DCS) becomes the following:

$$DCS = PNP + PVP + SP$$

By breaking a sentence into three components rather than two, the word order of the sentence becomes clear.

$$DCS = PNP\ (S) + PVP\ (V) + SP\ (O)$$

If a sentence is split into only two fragments– noun phrase (NP) and verb phrase (VP)–this information is not sufficient for finding the correct word order. However, with a three-component

characterization of a sentence, we can determine the word order as well as the sentence's classification.

Disregarding verbless expressions such as "Yes," "Happy birthday," and "What a game," PVP must be present in every sentence. But PNP or SP may or may not be necessary. This results in having the following types of sentences in English:

Type I: PNP + PVP + (SP)
Type II: PVP + (SP)
Type III: PVP + PNP + (SP)

Type I is declarative sentences (DCS). It can also be considered exclamatory sentences (ECS), declarative sentences that express strong emotions. Type II lacks PNP, meaning imperative sentences (IPS) in English. Commands and requests fall under this type. Type III has PNP and PVP in reverse, creating interrogative sentences (ITS) or questions. The four types can now be redefined as the following:

Declarative / Exclamatory: PNP + PVP + (SP)
Imperative: PVP + (SP)
Interrogative: PVP + PNP + (SP)

In some cases, exclamatory sentences such as rhetorical questions can take the form of PVP + PNP + (SP).

# 4 Synapper Models

## 4.1 The Merge of Word Orders

We divide the six word orders into two groups by looping them around. SOV, OVS, and VSO are one group whereas SVO, VOS, and OSV belong to another group. The only difference between the two groups is the direction of flow. If one group is assumed to flow in the clockwise direction (e.g. from S to V to O for the SVO word order), then the other group is assumed to flow counterclockwise. Connecting the first and the last constituents of a sentence creates a loop that can be applied to any language in any given word order. However, some sentences require some of the words to be linked in two or more dimensions or directions. If a word such as *blue* depends on the presence of another word such as *bird*, then the dependent word is linked to only the related word and not to the rest of the sentence. We define this approach as the *synapper*. The synapper is a mechanism that utilizes multiple dimensions in order to connect tokens such as words. In linguistics, it attempts to represent syntactic structures of sentences. By creating models of the synapper, translations of even complex sentences in different word orders can be merged into unified syntactic structures.

## 4.2 Synapper Modeling of Declarative Sentences

We use the following declarative sentence as an example to investigate whether its syntactic structure is uniform for English (SVO), French (SVO), Japanese (SOV), Uzbek (SOV), and Welsh (VSO):

Jane has a very fast brown horse.

It has the default PNP + PVP + (SP) arrangement in English, where the subpredicate is composed of determiner + adverb + adjective + adjective + noun:

DET + ADV + ADJ + ADJ + N

By arranging these words in more than one dimension, we can create the synapper model of the sentence. The words that belong to the main circuit are called *nodes*. Any word that is connected to a node from a different dimension is called a *branch*. A *constituent* is defined as a node with its branches.

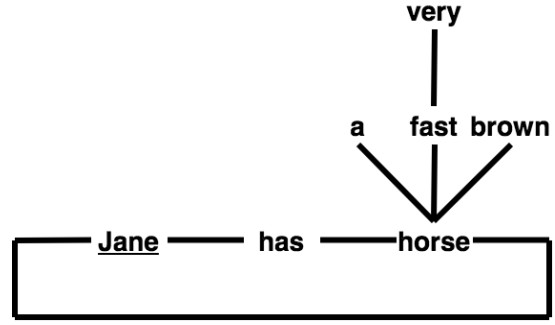

Figure 1: The starting constituent for English is underlined (Jane). In SVO languages, the sentence is read clockwise starting with PNP. The branch words that are connected to the node *horse* are read with the far-left word first (a, very, fast, brown). In some languages like French and Spanish, some branch words are supposed to be read after the node (a, horse, brown, very, fast).

Here is the breakdown of the way the words in the sentence are ordered in each language:

- English: Jane has a very fast brown horse.

- French: Jane has a horse brown very fast. (Jane a un cheval brun très rapide.)

- Japanese: Jane very fast brown horse has. (ジェーンはとても早い茶色の馬を持っている。)

- Uzbek: Jane very fast brown horse has. (Janeda bir juda tez jigarrang ot bor.)

- Welsh: Has Jane horse brown very fast. (Mae gan Jane geffyl brown cyflym iawn.)

Because SOV and VSO have the same direction of flow (e.g. counterclockwise), this sentence in Japanese, Uzbek, and Welsh should flow in the same direction. The only difference is Japanese and Uzbek start with the subject *Jane* where Welsh starts with the verb *has*. For English and French, the sentence is read in the opposite direction (e.g. clockwise) since SVO belongs to the other group along with VOS and OSV.

This means we can take the synapper model in Figure 1 and derive the perfect translation in each language. In other words, a single syntactic structure has all the sufficient information for expressing the same thought in any particular language as long as the word order and the direction of flow are known. For instance, this structure can yield the following sentence by traveling counterclockwise starting with PNP:

Jane very fast brown horse has.

Now we can simply replace the English words with Uzbek words and then morphemes can be added, changed, or removed such as the determiner *a* based on the language's grammar. The result is "Janeda bir juda tez jigarrang ot bor," which is the correct translation in Uzbek.

## 4.3 Synapper Modeling of Interrogative Sentences

Creating the synapper models of interrogative sentences requires a few more steps. Languages like English switch position of the subject (PNP) and the verb (PVP) to turn a declarative sentence into a question. However, this is not true at all for many other languages. They use verb conjugations or other methods to create interrogative sentences. If the function of language is to create thoughts, then the declarative form of a sentence becomes the default form. That means turning a declarative sentence into an interrogative style would require additional rules. These rules differ from language to language. So interrogative sentences must resort

back to their declarative forms for their synapper models to work with other languages.

Here is an interrogative sentence in English:

Why is Tim going to the hospital?

The sentence can become declarative by removing the word *why* from the sentence and then changing the word order to PNP + PVP + SP:

Tim is going to the hospital.

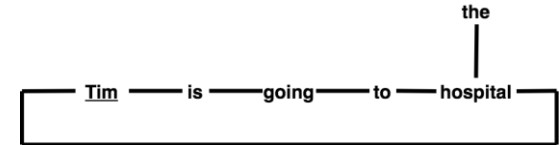

Figure 2: For SOV languages, the sentence is read counterclockwise starting with *Tim* (Tim, the, hospital, to, going, is).

Now the synapper model in Figure 2 can be applied to different languages:

- English: Tim is going to the hospital.

- French: Tim is going to the hospital. (Tim va à l'hôpital.)

- Japanese: Tim the hospital to going is. (ティムは病院に向かっている。)

- Uzbek: Tim the hospital to going is. (Tim kasalxonaga ketayapti.)

- Welsh: Is Tim going to the hospital. (Mae Tim yn mynd i'r ysbyty.)

To add the word *why*, different rules have to be applied. For English and French, the word is placed in the beginning of the sentence and then PNP and PVP are switched. In Japanese and Uzbek, the word is placed before *Tim*. In Welsh, it is put in the beginning of the sentence without moving PNP and PVP. So the interrogative forms become as follows:

- English: Why is Tim going to the hospital?

- French: Why is Tim going to the hospital? (Pourquoi Tim va-t-il à l'hôpital?)

- Japanese: Why Tim the hospital to going is? (なぜティムは病院に行っているのですか？)

- Uzbek: Why Tim the hospital to going is? (Nega Tim kasalxonaga ketayapti?)

- Welsh: Why is Tim going to the hospital? (Pam mae Tim yn mynd i'r ysbyty?)

Since interrogative sentences are essentially modified versions of declarative sentences, their grammatical rules are not necessarily identical between languages. If different languages have different rules of grammar to create interrogative sentences, then these rules must be implemented to synapper modeling accordingly one by one.

## 4.4    Recursion

One of the properties of language is its ability to be recursive. A recursive sentence can be made by adding phrases like *I think* or *It is true that*. Recursion enables varying degrees of complexity in sentences and thoughts. To model recursion in declarative sentences, some constituents have to be embedded or layered inside the main circuit. The following is a recursive sentence:

The fact that Colette was Willy was a big secret.

The first six words make up the primary noun phrase of the sentence. Because recursion is applied twice within PNP, a loop can be formed to *Colette was Willy* and then it can be looped again with the first three words of the sentence.

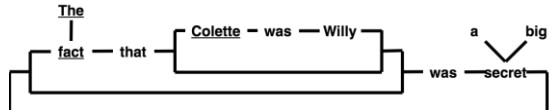

Figure 3: The primary noun phrase is in loops/layers of its own. They all have the same direction of flow (clockwise or counterclockwise) for each language.

The recursive layers travel in the same direction as the main loop, being consistent with the word order's direction of flow. Here is the correct arrangement in each language:

- English: The fact that Colette was Willy was a big secret.

- French: The fact that Colette was Willy was a big secret. (Le fait que Colette soit Willy était un grand secret.)

- Japanese: Colette Willy was that fact a big secret was. (コレットがウィリーだった事実は大きな秘密だった。)

- Uzbek: Colette Willy was that fact a big secret was. (Colettening Willy ekanligi fakti katta sir edi.)

- Welsh: Was the fact that Colette was Willy a big secret. (Roedd y ffaith mai Colette oedd Willy yn gyfrinach fawr.)

Although one syntactic structure accurately represents the sentence in all five languages, the starting point of the sentence can be different. The first word in Japanese and Uzbek is *Colette* whereas *the fact* are the first two words for English and French. In Welsh, the first constituent is *was* since Welsh is a VSO language. However, the direction of flow for the Welsh sentence is different from Japanese's and Uzbek's. In Figure 3, the sentence should be read clockwise for English and French and counterclockwise for the other three languages. But the Welsh translation behaves as if it is not actually a VSO sentence. Instead, the word order appears to be the same as English, SVO. The only difference is the verb is placed at the beginning of the sentence for Welsh. This phenomenon can be observed in Figure 2 as well. If Welsh is truly a VSO language, then the correct order of translation should be *Is Tim the hospital to going*. This would match the direction of flow of Japanese and Uzbek as it should. But the correct translation in Welsh is *Is Tim going to the hospital*. This is no different from the original sentence in English except for placement of the verb. Thus, based on the evidence, we find that Welsh's actual word order is not VSO. It appears to be VSO only because the verb is placed before the subject. However, it cannot be a VSO language since the direction of flow matches that of SVO. So Welsh's real word order seems to be SVO-V1. V1 or *verb-initial* indicates the verb must be placed before the subject and the object regardless of the word order.

## 4.5    Ambiguity

The concept of ambiguity raises an interesting question regarding whether the meaning of a sentence is actually morphed by its structure. An English speaker can easily tell the difference of a phrase *although he knew I told him* between *He was surprised, although he knew I told him* and *He was surprised. Although he knew, I told him*. In the first instance, the phrase behaves as a subordinate clause. In the second sentence, *although he knew* is a subordinate clause whereas *I told him* is the main clause. The same words are used in the exact same

order for representing two independent thoughts. Therefore, the synapper model for each expression should not be the same. The first sentence has *I told him* embedded in the structure *he knew X* where *X* is replaced by *I told him*. In the second expression, the subordinate clause a*lthough he knew* is simply inserted before the main clause, *I told him*, without any embedding. As the meaning of the expression changes, the syntactic structure also changes. In other words, the meaning changes as a sentence's syntactic structure changes.

We should note that a synapper model can have more than one meaning in some circumstances. If a word used in a sentence has more than one definition or if it belongs to more than one lexical category, the same structure can defer semantically. The word *orange* as in *Her answer was orange* can refer to a fruit or a color.

## 4.6   Comparisons of MT Models

We further examine the potential effectiveness of synapper modeling for MT by putting it to test with a complex Korean sentence. Then we compare the result with currently available machine translation services such as Bing Microsoft Translator, Google Translate, and Naver Papago.

The following is a sentence from a news article by Yonhap News, *불붙는 우주관광...베이조스 오는 20일 여행도 항공당국 승인* (Space travel heating up... Bezos also approved for travel on the upcoming 20th by the Federal Aviation Administration):

영국 억만장자 리처드 브랜슨 버진그룹 회장의 민간 우주 관광 시험비행이 성공하며 '스타워즈 시대'의 포문을 연 가운데 미 연방항공국(FAA)이 제프 베이조스 아마존 이사회 의장이 이끄는 블루 오리진의 유인 우주비행을 승인했다고 로이터 통신이 12 일(현지시간) 보도했다. (Kim, 2021).

This declarative sentence contains 35 words. An English translation by a human is as follows:

> Reuters News Agency reported on the 12th (local time) that the U.S. Federal Aviation Administration (FAA) approved manned space travel from Blue Origin, led by Jeff Bezos, the chairman of the Amazon Board, in the midst of opening the door to the 'Star Wars era' by succeeding a test flight for civilian space travel from a British billionaire, Richard Branson, the chairman of Virgin Group.

The 35 words in the original text has ballooned to 65 words for the English translation, an 85.7% increase. This is due to a couple of factors. First, the Korean language does not use articles such as *a/an* and *the*. So articles must be added to nouns in the English translation when applicable. Second, words or phrases such as *Federal Aviation Administration* in Korean are considered single units, making them essentially one word each. Third, Korean adjectives and verbs can be grouped together, which also reduces word count.

The complexity of this Korean sentence can be challenging for the current generation of machine translation software. Having a large number of words in a sentence can exponentially increase the number of translation possibilities for what MT might consider as correct. It also likely increases the chance of producing an error in the translation since the more the number of words a sentence has, the more the number of possibilities for error exists. In fact, Google Translate gave two different Korean-to-English translations for the exact same input in Korean, alternating between the two solutions when the service was accessed on different days. Here is one of the translations given by Google Translate:

> Google Translate, Version 1 (49 words):
> British billionaire Richard Branson, chairman of the Virgin Group, opened the 'Star Wars era' with a successful private space tourism test flight, and the Federal Aviation Administration (FAA) has approved Blue Origin's manned space flight, led by Amazon Board Chairman Jeff Bezos. Reuters reported on the 12th (local time).

In the original sentence, the subject–*Reuters News Agency*–was located toward the end. This is somewhat unusual for the Korean language since the default word order in Korean is SOV. But, because of the extremely lengthy subpredicate (30 words), the journalist decided to put the subject at the end of the sentence with the main verb. If the

algorithm used by Google Translate fails to locate the subject or PNP properly, the translation will likely result in error. In Version 1, the English translation has a different noun phrase as the subject with the word *opened* as the main verb, which is also incorrect. The translation placed the subject and the main verb of the original sentence into a separate sentence.

Google Translate, Version 2 (50 words):
> The U.S. Federal Aviation Administration (FAA) has approved Blue Origin's manned space flight, led by Amazon Board Chairman Jeff Bezos, as British billionaire Richard Branson, chairman of the Virgin Group, successfully test flights for private space tourism, ushering in the "Star Wars era" Reuters reported on the 12th (local time).

Version 2 correctly translates the source as one sentence. Overall, the translation holds the essence of the original text's message. However, the words *has approved* in the beginning of the sentence should simply be *approved* as in *approved on the 12th of July* since the news article is reporting what took place on a particular date. Also, because of the way the words are ordered, it is somewhat ambiguous whether Jeff Bezos led Blue Origin's manned space flight or that he led the U.S. Federal Aviation Administration (FAA). This confusion does not exist in the original sentence.

Bing Microsoft Translator (44 words):
> British billionaire Richard Branson's successful private space tourism test flight opened the door to the "Star Wars era," reuters reported on Thursday (local time) that the FEDERAL AVIATION ADMINISTRATION (FAA) had approved Blue Origin's manned space flight, led by Amazon Board Chairman Jeff Beizos.

Although this translation may be adequate for comprehension, it combines two different thoughts as one in the form of *A opened B, it reported that X had approved Y*. This might be due to the fact that the MT algorithm could not decipher what was actually reported by Reuters while still requiring the translation to be a single sentence. In addition, the word *Thursday* is not present in the Korean sentence but was added to the English translation

somehow. The date mentioned in the news article is supposed to be July 12, 2021, which is a Monday.

Naver Papago (42 words):
> The Federal Aviation Administration (FAA) has approved a manned space flight of the Blue Origin, led by Amazon Chairman Jeff Bezos, amid a successful private space tourism test flight by Virgin Group Chairman Richard Branson, Reuters reported on the 12th (local time).

Papago is a translation service from Naver Corporation, a company based in South Korea. The translation result is somewhat similar to Google Translate's (Version 2) in terms of its structure. However, it is missing an entire segment of the original text regarding the Star Wars era.

Synapper modeling of the same sentence takes a completely different approach. Here we shall address the fact that it does not technically translate sentences from one language to another in the traditional sense. Instead, synapper modeling constructs the correct syntactic structure of a sentence for all languages (language-independent) and then produces output in the targeted language (language-dependent). Since English's word order is SVO and Korean's word order is SOV, the words of the synapper model for the original sentence have to be read in the opposite order for English. However, because the journalist put the subject at the end of the sentence, it is no longer an SOV sentence. So the subject has to be moved to the beginning of the sentence to make the sentence's word order SOV. (Since the sentence is overly complex, the writer likely put PNP and PVP together at the end because SP became too lengthy.)

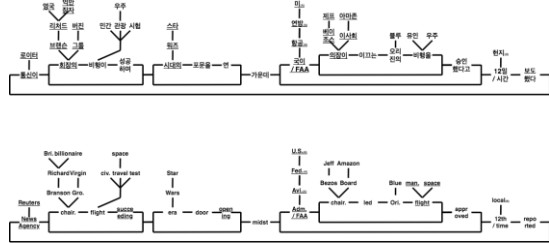

Figure 4: The syntactic structures are 100% identical for the two languages. (See Appendix A for an enlarged version of Figure 4.)

Once the words are changed from Korean to English, the synapper model can generate the correct English translation. To make the sentence

SVO as in English, it starts with PNP followed by PVP and then finishes with SP by traveling counterclockwise for all the loops present in the model. The following is the outcome:

> Reuters News Agency reported *on the* 12th (local time) *that the* U.S. Federal Aviation Administration (FAA) approved *a* manned space flight *from* Blue Origin led *by* Jeff Bezos, *the* Amazon Board chairman, *in the* midst *of* opening *the* door *to the* Star Wars era *by* succeeding *a* civilian space travel test flight *from a* British billionaire Richard Branson, *the* Virgin Group chairman.

43 words were derived from the synapper model. When articles and prepositions are added (as shown in italic), the total number of words in the sentence increases to 62, which nearly matches the 65 words in the human translation. Also, the output does not have any of the inaccuracies that were discussed in the five translation results from the four web services. This is likely due to using no probabilistic computations, which would cleave the sentence into parts and reassemble them for the output. By keeping the syntactic structure intact, any nuance or human element present in the source is much more likely to remain in translation.

## 5    Conclusion

The application of synapper modeling for machine translation has many advantages over today's predominant computation-driven approaches. By design, probabilistic models of machine translation such as SMT and NMT must use approximation for result. (Johnson et al., 2017). Although incremental changes can be applied to improve performance, the effect of diminishing returns will eventually pervade with time. The same phenomenon can be observed in other areas such as weather forecasting and board gaming. The amount of improvement that can be obtained is almost always greater in the initial stage of development than later. This is a limitation of taking probabilistic approaches. Synapper modeling, on the other hand, gets rid of this drawback significantly. We speculate that the human brain perhaps utilizes the same basic mechanism for the utilization of language such as translation and sentence formation. If so, implementation of this system in MT will likely improve the quality of machine translation to the level of human translators without requiring considerable computing power.

The theory of Universal Grammar (UG) also should be reexamined. We have demonstrated the possibility of syntax-semantics unity with synapper modeling. If the syntactic structure of a thought is identical for all natural languages, the assertions that language is innate and all natural languages are compatible with each other (Chomsky, 2000) could turn out to be true. Chomsky and several other linguists have long suspected that the grammars of various languages only differ in the setting of certain innate parameters among possible variants. (Carnie, 2002). Now we hypothesize that these parameters are simply the direction of flow and the starting point of a sentence, based on the word order of a language. However, since thousands of natural languages exist, more research should be conducted before we consider UG as a correct theory.

Parsing sentences linearly (e.g. from left to right) is too limited in scope to properly analyze their syntactic structures. When comparisons are drawn between different languages, it is especially apparent that one-dimensional representations are unproductive for NLP. By using multiple dimensions, on the other hand, it is possible to realize the uniform syntactic structure for each syntax-semantics entity for all natural languages. Perhaps this may not be such a surprising outcome considering Chomsky's long-held proposition that "linguists must be concerned with the problem of determining the fundamental underlying properties of successful grammars. The ultimate outcome of these investigations should be a theory of linguistic structure in which the descriptive devices utilized in particular grammars are presented and studied abstractly, with no specific reference to particular languages." (Chomsky, 2015).

## 6    Discussion

Due to limited resources, our research falls short on establishing a working MT system as a rule-based MT model. Further research should be done in collecting more data–qualitative and quantitative– as well as exploring synapper modeling with other areas of syntax theories such as ellipsis. Additionally, more research is desired in linguistics and neuroscience in order to verify the hypothesis on the utilization of multi-dimensional modeling mechanism used by the human brain for natural language processing.

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

## A   Figure 4 (Enlarged)

(See the next page)

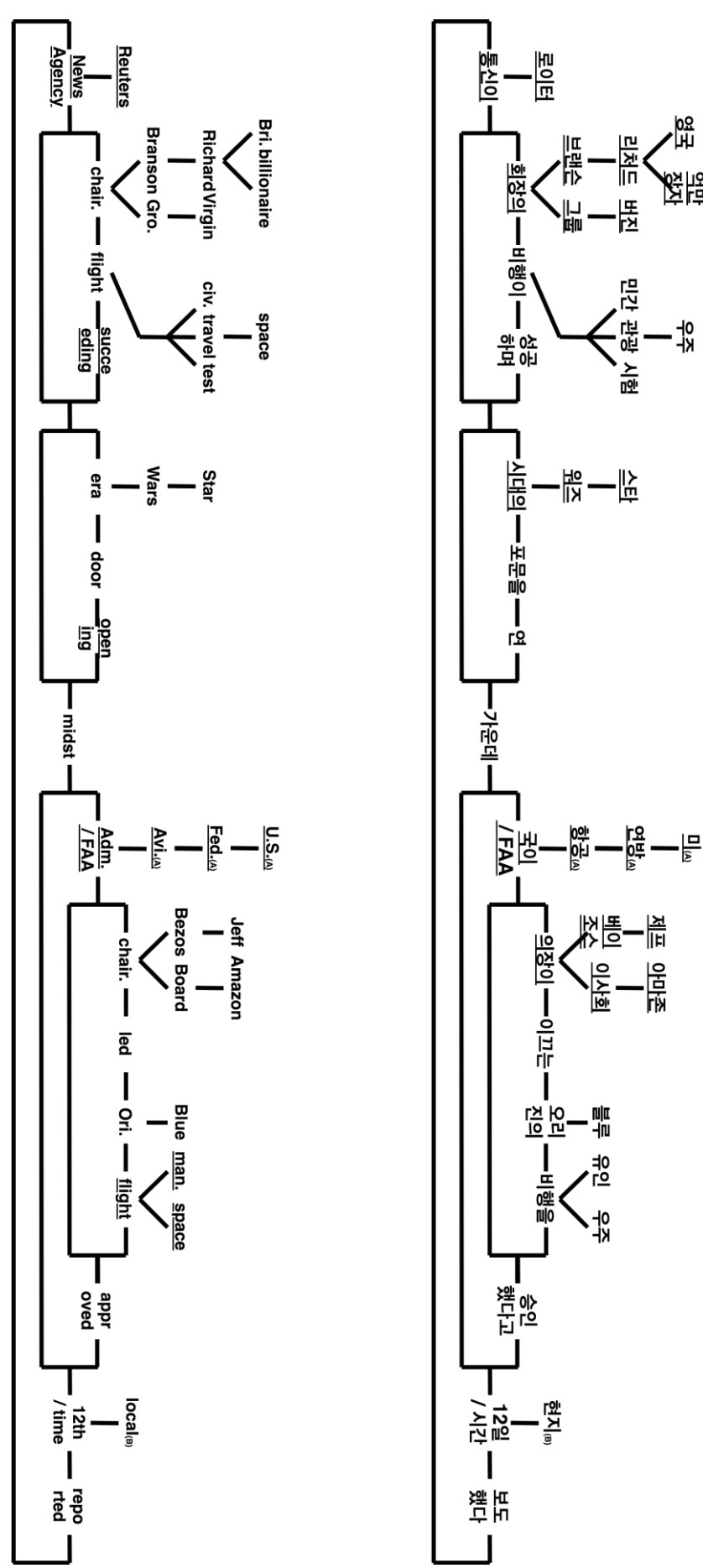

