# OpenReview forum: "The Uniformity of Syntactic Structures in Various Natural Languages"
_aclweb.org/ACL/2022/Workshop/CMCL — Submitted to CMCL 2022_

### Official Review · Reviewer_3nkC · 2022-03-20
**A bizarre submission**

**Rating:** 1
**Confidence:** 5

**Review:**

The paper is too general, too vague and full of statements that are not supported by references, experiments or even a rigorous case study. Closer to central theme of the paper, the denouncement that applications of syntax trees have been very limited in scope, with the exception of UD ignores a rich tradition of computational linguistics and NLP research. Even if we limit our scope to the field of MT, there are a number of studies that explicitly use syntactic parsing as part of their modeling which is not mentioned (let alone compared to) in this paper.

There are numerous misconceptions in this work, the two most pressing ones being the idea that there is such a thing as "100% accurate translation" (ignoring the fact that translation is highly subjective and even human expert translations can produce more than one perfectly valid translation), and that matching syntactic structures across languages is sufficient (or even necessary) for a faithful translation.

Finally (albeit less critical than it should be) there is the point that the paper is neither computational nor modeling (both of which are part of the workshop's name).

---

### Official Review · Reviewer_MJq5 · 2022-03-21
**An unclear approach to "Machine Translation" via cyclic graphs**

**Rating:** 1
**Confidence:** 5

**Review:**

I like to believe I always try to write constructive reviews, but I have to say that reading this paper was a puzzling experience.
As it is written, the paper lacks any depth in its relation to the vast variety of literatures the authors seems to want to connect to.
What is the paper about? Left to the introduction alone, one would first think it a critique of constituency-based approaches to syntactic structure. Then it seems to be about how purely syntactic approaches are unnecessary for the task of machine translation, then it becomes a critique of Probabilistic models (in general. Any model whose output is probabilistic is unfit for machine translation according to the authors).

The paper  introduces a representation of sentences in the forms of unlabelled, cyclic graphs: a curious chimera of constituency structure, dependency trees, and a hint to lexical-functional graphs --- that has none of the formal advantages of these existing approaches, while attempting to capture the same information. The authors claim this "new" representation is better for Machine Translation broadly defined, but there is no evidence of this from a quantitative nor qualitative perspective.

Content issues aside, the paper is just not thematically fit for CMCL. To clarify, I do think that there are many different ways to do “computational modeling”, and that a purely formal approach can be plenty “computational”. However, there is no modeling in here in any possible interpretation of the term, nor there is a computational contribution beyond the trivial existence of graphs (in fact, this is stated explicitly in the abstract!), nor there is anything resembling a “cognitive” topic.


## Specific Comments

Lack of focus aside, the paper makes an astounding number of unsupported claims (basically, one per paragraph), which are at best trivial and often plainly wrong. I address some of the broader ones in what follows to exemplify how the paper is completely disconnected from current work in linguistics and computational linguistics.

- The first part of the introduction seems to take issue with how constituency based approaches deal with the mapping between syntactic variation and semantic interpretation. While of course there are plenty of approaches that argue for a tighter relation between syntax and semantics, it is a strong claim that similarly interpreted sentences cross-linguistically should map to the same syntactic structure. Additionally, this shows a misunderstanding of what standard constituency approaches do --- that is exactly to give identical syntactic structures to the same grammatical relations across languages, modulo what the authors seem to care about most, *head directionality*. Even putting these two points aside, the issue of semantic representations across languages is one the paper never goes back to. In fact, the papers end up adopting a representation (in the form of cyclic undirected graphs) that has NO explicit encoding of basic grammatical dependencies.

- On the same page, the authors go ahead to claim *No one has yet been able to demonstrate how such a process can take place 6almost instantaneously in the human brain.*

    *Such a process* here being the process of establishing grammatical relations between words. Now, putting aside the decades of psycholinguistic work on parsing across a variety of syntactic frameworks that the authors seem to think not relevant, this is once again a point the paper never goes back to. Again, the “formalism ” adopted by the paper is then a form of cyclic graph  which the authors never bother to formally introduce. How are these graphs derived from sentences?

- We then get to what seems to be the main focus on the contribution: an improved machine translation model. Apart from not defining the scope of the problem at all, there is no attempt to contextualize the contribution with respect to the existing literature in the subfield --- apart from a few random references to transformers models. In fact, given how focused the intro is on syntactic representations, it is puzzling how the paper completely ignores the vast literature on how syntactic information (be if from constituency or dependency trees or both) has been incorporate in MT approaches of different kind.

- They also state that “probabilistic” approaches are problematic because they do not lead to a 100% accuracy — which I take to mean a 1-t-1 correspondence between two sentences in two different languages — which is of course a puzzling statement by itself given that contextual translation is never 1-to-1 even when done by human annotators.

- The rest of the paper becomes even harder to follow. The method section presents a “decomposition” of sentences based on S, V, O elements, which basically reintroduces constituency/dependency structures, again sprinkled with a variety of unsubstantiated claims.

    Crucially, their “model” is not, in fact, a model. There is no attempt to explain the properties of their representations, how the system is supposed to work theoretically, how it would handle things like modifiers alternation, and within language word order variation. The authors seem to believe this "multidimensional" approach to be somehow simpler that constituency/dependency trees, while of course formally it is far from being simple (and putting aside the fact that hierarchical representations are trivially multidimensional).

- There is also a puzzling section about “ambiguity” which is not about ambiguity at all. Their non-ambiguous examples aside, the authors quickly claim that lexical ambiguity can be dealt with within their “framework” (I use the term loosely) but do not even mention crucial cases of syntactic ambiguity?

- Additionally, it was truly unclear to me how to interpret the “Comparison of MT Models” section, as there is no comparison (unless we think extrapolating one paragraph from Google Translate counts as a comparison). Reading this section made it seem like the authors believe that translating a sentence can truly be reduced to the task of one to one translation of words (dismissed as an easy trick itself) and picking a point to start from in their graph.

---

### Official Review · Reviewer_DEGb · 2022-03-22
**The rediscovery of syntactic structure**

**Rating:** 1
**Confidence:** 5

**Review:**

This paper puts forward what I imagine is intended as a novel model of syntactic structure for natural language, which allows uniformities across languages to be straightforwardly characterized via a single underlying (declarative) structure. The paper then goes on to argue that this model could be used as the basis for an MT system.

Much as I am sympathetic to the importance of abstract structure in characterizing natural language syntax and in deriving underlying meaning, the current paper does not advance our state of understanding of this area. The proposed theory comes across as a rehashing of existing ideas that have popped up repeatedly in the history of grammatical description. The synapper structures for noun phrases look very much like dependency trees. The proposal for interrogatives in section 4.3 recalls Zelig Harris's idea that non-canonical sentence types need to be transformed into a "normal form".  The idea that structures can exhibit ambiguity goes back to Chomsky's earlier arguments (in "Syntactic Structures"). And the proposal that syntactic structure should allow recursive embedding has a similarly long history.  The one novel contribution concerns the cyclic character of word order patterns, and the division into clockwise and counter-clockwise languages. This is a novel idea, so far as I am aware (though one could imagine deriving this distinction from the approach to word order explored by R. Kayne in his "Antisymmetry of Syntax" book). This idea would be more compelling if it could be shown that the distinction between these two classes of languages correlated with some other property. Otherwise, it comes across as merely taxonomic.

The paper's discussion of MT is even weaker. After a detailed recounting of the performance of existing systems on the translation of a single Korean sentence into English, the paper puts forward what would be the translation of the same sentence derived from the current approach. The problem is, as far as I can tell, there is no implemented system. And in carrying out this translation, the paper permits arbitrary changes to be made in service of producing the translation (e.g. lines 596-600). Saying what the answer *would* be if such a system existed will not be convincing at all to readers with any appreciation of the complexity of building MT systems. While I don't think that NLP progress should be held hostage to performance on benchmark datasets, as incremental advances won't necessarily lead to fundamental progress, the bar for progress is higher than what is done here.

Finally, the paper presents this structure-based approach to MT as though it is a novel contribution. However, the idea that translation should be done via a mapping between a cross-linguistically uniform underlying structure is a well-studied one, called either syntactic transfer or interlingua, depending on the abstraction of the analysis.  For more recent syntactic-transfer approaches, see the discussion in Williams, Sennrich, Post, and Koehn's book on Syntax-Based Statistical Machine Translation. And it would be worth paying attention to the many well-studied cases of structural divergence, where the mapping from the structure of one language to the structure of another is far from isomorphism, since they will pose problems for the approach proposed in this paper. (For an old, but still useful discussion, see B. Dorr's 1994 Computational Linguistics paper "Machine Translation Divergences: A Formal Description and Proposed Solution", but there's lots of work since then on this topic.)  I should also point out that much of the debate on per-NN syntax-based MT models concerned how to best represent the mapping between languages so as to best accommodate the divergences that exist.

---

### Decision · Program_Chairs · 2022-03-29

Reject